# Psychoeducation Reduces Alexithymia and Modulates Anger Expression in a School Setting

**DOI:** 10.3390/children9091418

**Published:** 2022-09-19

**Authors:** Salvatore Iuso, Melania Severo, Antonio Ventriglio, Antonello Bellomo, Pierpaolo Limone, Annamaria Petito

**Affiliations:** 1Department of Clinical and Experimental Medicine, University of Foggia, 71121 Foggia, Italy; 2Department of Humanistic Studies, University of Foggia, 71121 Foggia, Italy

**Keywords:** psychoeducation, alexithymia, anger, bullying, school setting

## Abstract

Bullying and violence are relevant issues in school settings and negatively impact students’ well-being and mental health. Psychoeducation and anti-bullying programs may prevent violence among students by addressing emotional expression and regulation, alexithymia, and anger. We describe the impact of a psychoeducational intervention delivered to 90 male and 101 female school youths (N = 191), aged 12–14 years old, and aimed to improve their emotional recognition and regulation, as well as to reduce alexithymia in order to prevent aggression and bullying episodes. A psychological assessment has been performed before (T0) and after (T1) the intervention including levels of alexithymia, measured with the Toronto Alexithymia Scale-20 (TAS-20), the State-Trait Anger Expression Inventory (STAXI), Empathy Quotient (EQ), and the Emotion Regulation Questionnaire (ERQ). Females have shown higher levels of alexithymia at baseline whereas other characteristics (anger, empathy quotient and emotional regulation) did not differ among sex groups. The psychoeducational program significantly increased the empathy quotient (+10.2%), the emotional regulation reappraisal (+20.3%), and the assertive anger expression (+10.9%); alexithymia significantly decreased after the intervention in all the samples (−14.4%), above all among students scoring ≥61 at TAS-20 (−48.2%). Limitations include a small sample from a single school setting, the lack of a control group without psychoeducation, and an assessment based on self-reported measures. We may conclude that psychoeducation has significantly reduced levels of alexithymia and improved empathy and emotional regulation among adolescents.

## 1. Introduction

Bullying in school settings is a relevant socio-cultural and educational issue severely impacting adolescents’ well-being and mental health; anti-bullying programs are an urgent priority, and their effectiveness should be assessed in the real world and scientific literature [1]. Fraguas et al. [1] published findings of a large meta-analysis of randomized clinical trials (n = 69) including school anti-bullying interventions and concluded that they were efficacious in reducing bullying episodes and improving psychological health among students regardless their specific duration. In addition, the effectiveness of anti-bullying programs examined did not show any decrease over time. It has been argued that a set of skills are impaired in bullying: interpersonal skills, such as empathy and moral disengagement, coping strategies and management of stress, emotional expression and regulation [2]. Recently, Sen Demirdogen et al. [3] have reported that bullying among Turkish adolescents was significantly associated with lower levels of “reading minds in the eyes”, alexithymia traits as well as lower empathy. In particular, alexithymia, emotional components of alexithymia, as well as anger-expression styles, have been described as associated factors to bullying, cyberbullying, victimization and their perpetration [4,5]. Wachs and Wright [6] in their study involving 1590 adolescents found that behaviors of traditional bullying, cyberbullying, and the combination of the two modalities were all associated with higher levels of alexithymia, with greater disadvantage for the latter group. The authors argued that adolescents who are unable to recognize or describe their emotions report higher difficulty in modulating emotions as well as higher aggression toward others. Shabahang et al. [7], in a large study involving 250 students in Iran, found that factors strongly predicting bullying among adolescents included difficulty in engaging in goal-directed behaviors, non-acceptance at an emotional level, lack of emotional clarity, difficulties in impulse control, and poor emotional awareness. In 2019, Bakan et al. [8] conducted a psychoeducational program aimed to increase emotional competence among adolescents in a school setting with 90-min sessions per week over 9 weeks. They found that at the post-intervention assessment, students showed a decreased score on violence and alexithymia scales. Similarly, in 2022 La Grutta et al. [9] conducted a psychoeducational program among 229 students aimed to increase children’s emotional competence and scholastic skills. They found that students reported positive changes in emotional recognition after the intervention with better levels of emotional health and psychological well-being and a possible future positive impact on cooperative skills and greater school success.

According to the Toronto Group statement, alexithymia is no longer simply defined as the inability to recognize and communicate one’s emotions, but it is considered as a deficit in the cognitive processing of emotions, characterized by a bias in identifying and describing emotions both by concrete and as externally oriented thinking [10]. Alexithymia is also a personality dimension that increases susceptibility to several disorders related to the emotional dysregulation [10,11,12].

Specifically, the relationship between alexithymia and anger is largely debated. Berenbaum and Irvin [13] found that individuals with alexithymia (both men and women) displayed greater non-verbal anger responses during the execution of an anger-provocation task. These authors showed a significant association between alexithymia and anger as well as poor communication in individuals reporting high levels of alexithymia. Moreover, alexithymia is positively associated with aggression and some research evidence suggest that impulsivity may be a key factor mediating the relationship between alexithymia and aggression [14,15].

Increasing evidence supports the effectiveness of youth programs in reducing alexithymia, improving the ability to recognize and express emotions [6,7,8,9,16,17,18,19].

In this research report, we describe the impact of a repeated, systematic, prospective psychoeducational intervention delivered to 90 males and 101 females school youths (N = 191), aged 12–14 years old, and aimed to improve their emotional recognition and regulation as well as to reduce alexithymia in order to prevent aggression, anger dyscontrol, and bullying episodes.

## 2. Methods

### 2.1. Sample

Following three episodes of traditional bullying and homophobic bullying that occurred in the previous school year at Secondary School “L. Murialdo” in Foggia (Italy), we delivered this psycho-educational intervention in the framework of the Plan for Health Promotion and Education approved by the Region of Apulia (Italy). Preliminarily, we carried out awareness-raising activities aimed at students’ families in this urban middle school included in the public school district of Foggia (Italy), characterized by a medium socio-cultural level. Preliminary activities led to an increase of families’ level of information about bullying and anger expression among adolescents. After this informative step, students whose parents had provided consent (all students involved) for participation, were recruited into the psychoeducational activity. A systematized, repeated psychoeducation has been delivered to 90 males and 101 females (N = 191) school youths, aged 12–14 years old, attending a secondary school in Foggia (Italy), which were recruited and assessed at baseline (T0) and 2 months (T1) after the intervention (see the following section) aimed to improve their own emotional regulation and reduce bullying and violence. The repeated assessment (at T0 and T1) included the empathy quotient, alexithymia, anger and emotional regulation. Figure 1 shows the CONSORT flow diagram.

### 2.2. Psycho-Educational Intervention

This psychoeducational intervention has been delivered by the Unit of Clinical Psychology at the University of Foggia, Italy, as described in detail in the Catalogue of the Regione Puglia, 2018/2019 school year, and developed according to the framework of the Strategic Plan for Health Promotion and Education. It has been approved by Regione Puglia and has been designed for students at their third year course of a secondary school. The psychoeducation included different modules (delivered in eight meetings) aimed to promote personal awareness of emotions and the ability to make responsible choices: (a) emotional literacy; (b) sexuality; (c) relational communication skills; (d) bullying and homophobia; (e) prevention of violence. The project involved teachers, psychologists/psychiatrists, gynecologists and social workers from the University of Foggia. These modules have been inspired by a cognitive-behavioral approach on assertiveness and recognition of emotions [20] as well as interventions for the prevention of bullying and homophobia [21,22]. The meetings have also been based on teaching activities such as circle time, role-playing, and filmed images inspired by the Socratic method (*maieutics*), based on an active learning leading to the enjoyment of new awareness and direct participation of all students involved. Circle time is a group-based activity including youths from a school setting in an interactive manner. This technique, coordinated by expert and well-trained teachers/psychologists, provides time for listening trainers and peers, promoting interactive communication and learning new concepts and skills as shared in the group (e.g., emotions regulation and expression, how to manage anger, consequences of violence and bullying). Moreover, it promotes sensory experiences, socialization, and time for fun. The role-playing consists in changing behaviors, in an educational setting, in order to assume a new adopted role for learning new skills through a role-based experience. Scenes of anger, violence, and bullying have been simulated and students have been involved in the role of both aggressor and victim in order to understand how to manage the emotions in practice. We also employed filmed images and selected movies regarding emotions, anger, violence and bullying in order to potentiate students’ appraisal through practical examples as filmed in a real-life setting. All these techniques have been combined and employed in six meetings, focused on six different tasks, each lasting 3 h.

Meetings and modules are briefly described as follows:(a)Emotional literacy. This focused on elements of emotional education, including students’ acquaintance with each other, explanation about the training and sessions, training techniques about emotions. Exercises were also focused on producing situations based on anger emotion as well as neutral emotions in a sitting mode. Emotion has been produced through components of facial expression, breathing pattern alone, body posture, and emotion production using all of the three components [20].(b)Sexuality. A group psychoeducational intervention was carried out on the differences between sexuality, affectivity and feelings. In addition, the psychoeducational intervention deepened knowledge about sexually transmitted diseases, risk behaviors, and protective behaviors [23].(c)Relational communication skills. Relational communication skills were implemented through an assertiveness training, focused on the following communication skills: self-knowledge, listening to one’s own emotions, willingness and ability to show oneself and relate to the other in a respectful manner, express pleasant feelings, and make a request in a positive manner [20].(d)Bullying and homophobia. Bullying and homophobia were addressed in two steps. The first phase was focused on formation regarding sexual identity with special attention to its components such as biological sex, gender identity formation, transsexualism and transgenderism, and sexual orientation. A second phase was based on discussion and reflection regarding the phenomena of homophobia and transphobia with reference to stereotypes, prejudice, discrimination, and social representations [24].(e)Prevention of violence. This meeting was focused on the awareness and recognition of various types of violence and gender-based violence: physical, sexual, psychological, economic, workplace, and witnessing violence with reference to stalking and femicide. Risk indicators have been also discussed, such as systematic control of the victim, isolation of the victim, devaluing the victim by undermining self-esteem, victim submission, and victim conditioning [25].

### 2.3. Ethical Issues

Students’ parents were involved and provided their informed consent for recruiting school youths in the program: information has been delivered through the school website and a teaching meeting between parents, students, and principal investigators has been organized. The ethical approval has been obtained after the agreement between the Didactic Scientific Committee of the Secondary School “L. Murialdo” in Foggia, the Department of Clinical and Experimental Medicine of University of Foggia, and the Socio-cultural Association “Donne in Rete” operating in the city of Foggia (Prot. 0000157/U del 12 January 2019 12:31:57 IV.1—Piano dell’Offerta Formativa POF). This study has been conducted according to the Code of Ethics of the World Medical Association (Declaration of Helsinki) for studies involving patients and humans. This manuscript meets the Recommendations for the Conduct, Reporting, Editing and Publication of Scholarly Work in Medical Journals.

### 2.4. Assessments

Assessments were performed at baseline (T0) and two months (T1), before and after the intervention, respectively, in order to describe the impact of the psychoeducation on the selected emotional variables. We employed standardized tools with training in their use (experienced psychologists [AP, SI, MS] with levels of inter-rater agreement [κ-statistic ≥ 0.94]). The following instruments have been selected as appropriate and evidence-based.

Empathy Quotient (EQ; [26]). Developed in order to measure empathy, it consists of 60 items, 40 items assessing empathy and 20 control items. Each item may be scored “2, 1, or 0”, and rated as “strongly agree”, “slightly agree”, “slightly disagree”, or “strongly disagree”. The total score ranges from 0 (being the least empathetic possible) to 80 (being the most empathetic possible).

The Emotion Regulation Questionnaire (ERQ; [27]). This is a 10-item self-report questionnaire composed by two scales describing different emotion-regulation strategies: Cognitive Reappraisal (6 items) and Expressive Suppression (4 items). Each item is rated on a 7-point-likert scale from “strongly disagree” to “strongly agree”. We employed the Italian translation and validation of the ERQ by Balzarotti et al. [28].

The State-trait Anger Expression Inventory (STAXI; [29]). This tool explores the two major components of anger: (a) state anger (S-Rab, alpha coefficient for adolescent males 0.87, alpha coefficient for adolescent females 0.90), defined as an emotional state of varying intensity; (b) trait anger (alpha coefficient for adolescent males 0.82, alpha coefficient for adolescent females 0.84), defined as the disposition to respond to external situations with anger. This characteristic is explored in two subscales: (a) temperament led to anger (T-Rab/T; alpha coefficient for adolescent 0.85; 4 items) measuring the general predisposition to feelings of anger without a specific reason; (b) anger reaction (T-Rab/R; alpha coefficient for adolescent males 0.65, alpha coefficient for adolescent females 0.70; 4 items) reporting on differences in expressing anger when exposed to external stimuli. The concept of anger expression includes three main components: (a) anger out (AX/Out; AX/Out, alpha coefficient for adolescent males 0.73, alpha coefficient adolescent females 0.75; 8 items), anger toward other people or objects; (b) anger in (AX/In; alpha coefficient for adolescent males 0.84, alpha coefficient for adolescent females 0.81; 8 items), anger directed inward; (c) anger control (AX/Con; alpha coefficient for adolescent males 0.85, alpha coefficient for adolescent females 0.84; 8 items), individual difference in controlling the feeling of anger. A 24-item research scale has been derived from the sum of AX/Out, AX/In, and AX/Con subscales called expression of anger (AX/EX). The STAXI total score is base on 44 items, and each individual ranks his or her feelings of anger on 4-point scales that assess how it is expressed, repressed, and controlled.

The Toronto Alexithymia Scale, self-report 20-items (TAS-20; [30]). This scores alexithymia on a 5-point Likert scale, ranging from 1 (strongly disagree) to 5 (strongly agree), with a total score ranging from 20 to 100. The factor analysis of the TAS-20 suggests three main factors: (a) difficulty in identifying emotions (DIF); (b) difficulty in describing emotions (DDF); and (c) externally oriented thinking (EOT). The total TAS-20 score ≥61 is considered for identifying significant levels of alexithymia [8]. It is of interest that TAS-20 has been validated in both clinical and non-clinical samples [31,32,33]. Reliability: TAS-20 demonstrates a good internal consistency (Cronbach’s alpha = 0.81) and test–retest reliability (0.77, *p* < 0.01).

### 2.5. Statistical Analysis

Analyses were conducted by using Grand Prism 5 statistical software (San Diego, CA, USA). Means and standard deviations (SDs) were calculated for each characteristic and parameter, and results were considered statistically significant with two-tailed *p* < 0.05. Differences in psychometric dimensions between groups were compared by using the nonparametric Kruskal–Wallis test with Dunn’s post hoc multiple comparison test. Assessments of the relationship between TAS-20 and psychological dimensions (STAXI, EQ, Emotive Regulation Questionnaire) were performed by using Pearson’s correlation.

## 3. Results

We recruited 90 male and 101 female school-youths (N = 191), aged 13.2 ± 0.44 and 12.8 ± 0.03 years old, respectively. All students were involved in the psychoeducational program and assessed at T0 and T1.

Baseline characteristics (T0) are reported in Table 1. Psychological dimensions (STAXI, EQ, Emotional Regulation Questionnaire—ERQ) have shown no significant differences among males and females. Levels of alexithymia were higher among females at baseline with TAS-20 scoring of 58.0 ± 11.3 vs. 54.1 ± 10.2 among males (*p* = 0.0173).

The impact of psychoeducation on these variables has been measured at T1 as “%-changes” over time. Scores of STAXI state-anger increased by +0.70% (*p* = 0.0486) in general, but in particular among females (+1.41%, not shown), as well as Temperament Led to Anger increased by +14.4% (T-Rab/t, *p* = 0.0055), Anger Out by +10.9% (AX/Out, *p* = 0.0465), Empathy Quotient by +10.2% (EQ; *p* = 0.0092), all in both the sex groups. Alexithymia significantly decreased after the intervention by −14.4% in all the sample (TAS-20, *p* = 0.0001) as well as Emotional Regulation Suppression by −25.6% (*p* = 0.0007) whereas Emotional Regulation Reappraisal increased by +20.3% (*p* < 0.0001).

A significant positive correlation has been observed between alexithymia (TAS-20) and STAXI S-Rab (State Anger) as well as STAXI T-Rab (Trait Anger) at baseline (T0): this evidence may suggest that anger expression was somewhat influenced by students’ levels of alexithymia (Table 2).

In contrast, there was no significant correlation between psychological dimensions and alexithymia after the intervention, in particular anger expression was apparently no longer influenced by alexithymia, which significantly decreased across the sample (Table 2).

In addition, an effective reduction of alexithymia has been confirmed among those students (both males and females) which reported a clinically significant score at TAS-20, defined as ≥61: psychoeducation has strongly reduced alexithymia by −48.2% in this specific subgroup (*p* < 0.0001) (Table 3).

## 4. Discussion 

It has been largely described in the literature that bullying is related to personality traits such as aggression [34,35]. Alexithymia may increase emotional dysregulation [10,11] as well as verbal aggression, anger, and hostility even among adolescents [36]. Seidler et al. [37] have pointed out that emotions of shame can be expressed through anger in men and that alexithymia may reinforce this association. This study aimed to describe and test the impact of a repeated, systematic, prospective psychoeducational intervention aimed at improving emotional recognition and regulation as well as reducing alexithymia in a population of adolescents aged 12–14 years in order to prevent episodes of aggression and bullying. In fact, a great deal of evidence reports that anger and alexithymia are interconnected. Bermond et al. [38] provided neuro-functional evidence that anger and alexithymia are both underpinned by: (a) a deficit in the inter-hemispheric communication mediated by the corpus callosum; (b) excessive and dysfunctional activity of the right hemisphere; and (c) functional alterations of the prefrontal and anterior cingulated-cortex in the assessment of emotionally significant input. In fact, in our sample of students the anger-expression was somewhat influenced by students’ levels of alexithymia at baseline, including state- and trait-anger. A pioneering study by Joukamaa and colleagues [39] on 6000 adolescents (mean age 15–16 years) reported a prevalence of alexithymia of 9.5% among girls and 6.9% among boys. Similarly, we also detected higher levels of alexithymia among females at baseline (TAS-20: 58.0 ± 11.3 in females vs. 54.1 ± 10.2 in males; Table 1) and 38.6% of female students reported a significant clinically relevant alexithymia vs. 22.2% of male subjects (Table 3).

Some training programs based on the emotional-expression as well as psychoeducational interventions have led to a significant reduction of alexithymia in students’ samples [8,18]. Moreover, self-reported difficulties with emotional regulation, awareness of emotions, and levels of alexithymia, assessed by the 20-item Toronto Alexithymia Scale (TAS-20), were significantly reduced after an appropriate skill training session for adolescents [16]. Moreover, psychoeducation and emotion regulation skills-training appear to be more effective among adolescents than in adults, suggesting that there is greater range for improvement at a younger age [8,18]. In our study, the psychoeducational intervention has effectively reduced alexithymia across the sample (−14.4% of TAS-20 scoring at T1; Table 1), even among those students reporting a clinically significant alexithymia (−48.2% among students with TAS-20 ≥ 61 at T0). These findings fully confirm the efficacy of the delivered intervention in this study. In addition, our results confirm the emerging evidence that psychoeducation aimed to reduce alexithymia has a positive impact on emotional regulation and expression. In our sample, the Empathy Quotient increased as well as the Emotional Regulation Reappraisal. Conversely, emotional regulation suppression has been reduced by the intervention with a following increase of state and temperamental anger expression in an assertive manner. This assertiveness has been confirmed by the finding of a lack of association between anger expression and (reduced) alexithymic levels at T1. In fact, it has been described that people with high levels of alexithymia show lower ability to mentalize emotions and express them in an assertive manner [11]. Although alexithymia can be considered as a cognitive deficit of processing emotional aspects, while mentalization as a function extended to all types of mental states (e.g., emotions, desires, intentions and beliefs), according to Taylor [11], we can suggest a partial overlap between the construct of alexithymia and mentalization [40]. Alexithymia is considered a personality dimension, whereas mentalization is a mental activity. It has been described that people with high levels of alexithymia show lower mentalization of emotions [11]. College students with alexithymic traits performed worse on “theory of mind” tasks and showed less activation of the medial prefrontal cortex, which is an important area for mentalization processes [41]. Furthermore, some evidence suggests that alexithymia is associated with a reduced propensity to engage in the “theory of mind” [42]. We may speculate that our intervention has improved the access to mental states by promoting the recognition and expression of emotions, as suggested by the reduction in levels of alexithymia.

In Italy in 2016, Nocentini and Menesin [43] conducted a specific anti-bulling program named KiVa (Klusaamista Vastaan, literally “Against Bullying”) among 2041 students involving grades 4 and 6 and characterized by an intervention vs. control group. The KiVa program led to a reduction of victimization and bullying as well as increased pro-victim attitudes and empathy, above all among grade-4 students receiving the intervention. Even if our study shows a different design and more methodological limitations (number of subjects and lack of a control group), findings were somewhat similar in showing an improvement of anger management and emotional attitudes. In 2019, Costantino et al. [44] published findings from a two-year study conducted in 2017–2019 (BIAS study: The Bullying in Sicilian Schools) involving nine institutions in Palermo with a pre- and post-intervention assessment based on a questionnaire investigating verbal and physical bullying alongside resources of pro-sociality and resilience among 402 students. They concluded that a decrease in the number of bullying episodes has been reported over time. These results are interesting, and our study may be similarly completed with a long-term follow-up of bullying episodes in order to evaluate the effectiveness of our intervention. More recently, in 2022, another Italian research group [45] tested the effectiveness of the adapted version of “Equipping Youth to Help One Another (EQUIP) for Educators” (EfE) program. A pre- post-test design assessment with a control group involving 354 Italian middle and high school children has shown that males from the experimental group in particular have shown a significant disengagement in bullying after the program. Interestingly, in our sample, there were no differences in anger-state traits at baseline among sexes even if alexithymia levels were higher among females. Anger and alexithymia improved in both sexes after the intervention.

The improvement in emotional expression and regulation (including anger) and the reduction of alexithymia among students may be considered as favorable preconditions leading to lower rates of bullying and violence in the school setting. In fact, future studies might describe the long-term effect of such intervention in terms of bullying episodes and violence. In addition, it might describe the impact of alexithymia-focused interventions in improving students’ mentalization skills as well as emotional regulation and their contribution to a variety of interpersonal problems in adolescence.

Implications of our findings may include an improvement of emotional recognition, expression and regulation among adolescents leading to a higher subjective psychological well-being, a reduction of anger episodes as well as violence (including bullying) inside and outside the educational environment, an improvement in the classroom climate with positive impact on learning and school performances, the promotion of routine psycho-educational interventions in the school setting, and awareness campaigns against violence, bullying, and other behavioral disorders among adolescents. It would be of interest to develop future follow-up descriptive studies to measure these possible implications over time.

Limitations may include the small number of students involved from a single school setting, and the psychoeducational intervention might have influenced implicit biases and cognitive distortions about emotional and assertive skills, but not necessarily adaptive behaviors that need long-term follow-up. In addition, the cause-and-effect relationships involved in the observed changes are not confirmed by a control condition lacking psychoeducation (control group without psychoeducation intervention). This limitation suggests future comparison studies. In addition, assessment tools based on self-reported measures have been employed.

## 5. Conclusions

In conclusion, psychoeducational programs aimed at reducing alexithymia and improving the regulation of emotions, including anger, might be considered as a possible favorable intervention for preventing bullying and violence in school settings. The findings of this study suggest that, beyond limitations, our approach may be considered as a possible base for developing tailored interventions aimed to improve emotional awareness and reduce violence in youths.

## Figures and Tables

**Figure 1 children-09-01418-f001:**
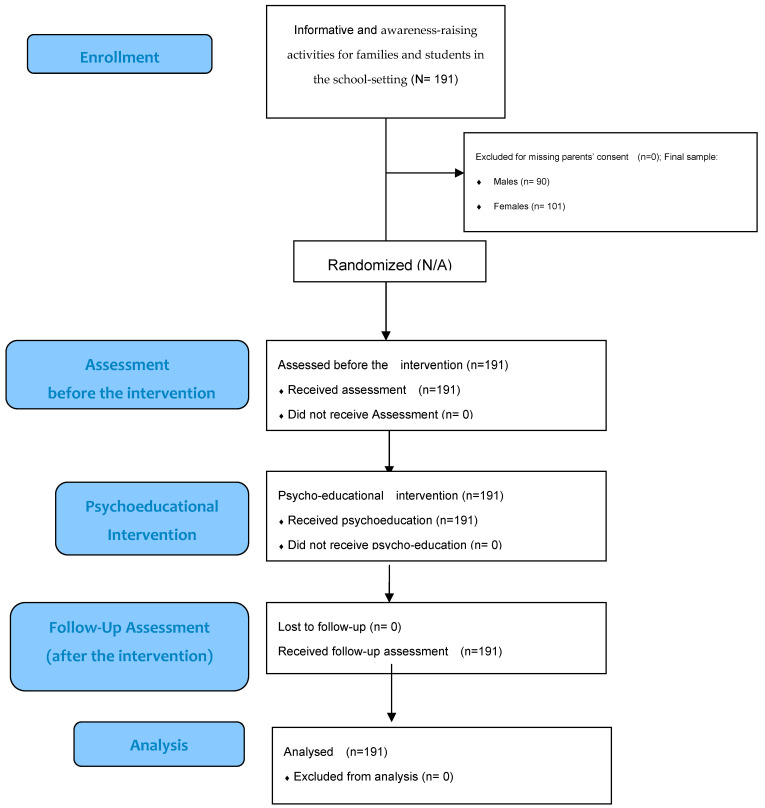
Psycho-educational intervention in the school setting: enrollment, assessment, intervention, follow-up, and analysis [CONSORT (2010) flow diagram].

**Table 1 children-09-01418-t001:** Differences in psychological dimensions before (T0) and after the psychoeducational intervention (T1).

Measures	T0				T1			*t*	Changes (%) Total (ΔT1-T0)	(Changes) *p*-Values
	Males (n = 90) (Mean ± SD)	Females (n = 101) (Mean ± SD)	Total (n = 191) (Mean ± SD)	*p*-Values	Males (n = 90) (Mean ± SD)	Females (n = 101) (Mean ± SD)	Total (n = 191) (Mean ± SD)			
**STAXI** *S-Rab*	**14.6 ± 4.37**	**13.9 ± 4.51**	14.2 ± 0.32	0.3075	14.5 ± 6.06	14.1 ± 6.49	14.3 ± 0.45	7.87	+0.70	**0.0486**
**STAXI** *T-Rab*	23.0 ± 5.26	23.6 ± 5.88	23.3 ± 0.40	0.5053	21.7 ± 7.41	23.1 ± 7.17	22.4 ± 0.53	2.21	−4.01	0.529
**STAXI** *T-Rab/t*	7.85 ± 2.62	8.27 ± 2.64	8.07 ± 0.19	0.3122	8.71 ± 3.96	10.2 ± 4.89	9.40 ± 0.32	12.60	+14.4	**0.0055**
**STAXI** *T-Rab/r*	11.3 ± 3.66	11.0 ± 3.19	11.1 ± 0.24	0.6414	11.5 ± 3.75	12.6 ± 4.20	12.0 ± 0.29	7.11	+7.50	0.0685
**STAXI** *AX/In*	17.1 ± 5.91	16.7 ± 4.92	16.9 ± 0.39	0.6033	16.9 ± 5.42	18.1 ± 4.98	17.4 ± 0.38	4.35	+2.87	0.2253
**STAXI** *AX/Out*	17.0 ± 4.48	17.0 ± 4.08	17.0 ± 0.30	1.0000	18.2 ± 6.36	20.2 ± 8.46	19.1 ± 0.54	7.97	+10.9	**0.0465**
**STAXI** *AX/Contr*	20.5 ± 4.64	20.9 ± 5.33	20.7 ± 0.36	0.6189	22.8 ± 8.23	23.1 ± 8.78	23.0 ± 0.62	4.37	+10.0	0.2239
**STAXI** *AX/EX*	29.2 ± 12.3	28.6 ± 10.6	28.9 ± 0.82	0.7726	29.7 ± 15.6	30.1 ± 11.8	29.8 ± 1.02	1.13	+3.02	0.7684
**EQ**	35.1 ± 10.5	38.3 ± 11.8	36.8 ± 0.81	0.0523	41.0 ± 20.3	40.7 ± 10.0	41.0 ± 1.20	11.5	+10.2	**0.0092**
**TAS-20**	54.1 ±10.2	58.0 ± 11.3	56.2 ± 0.75	**0.0173**	48.2 ± 15.5	50.1 ± 15.5	49.1 ± 1.10	20.30	−14.4	**0.0001**
**ERQ** *Reappresal*	23.8 ± 8.12	24.8 ± 9.21	24.3 ± 0.63	0.4375	29.8 ± 10.7	31.4 ± 11.0	30.5 ± 0.79	27.90	+20.3	**<0.0001**
**ERQ** *Suppression*	18.0 ± 7.9	18.9 ± 8.20	18.6 ± 0.58	0.4746	14.8 ± 5.91	14.8 ± 4.38	14.8 ± 0.38	22.30	−25.6	**0.0007**

Footnote: STAXI, state-trait anger expression inventory; S-Rab, state anger; T-Rab, trait anger; T-Rab/t, temperament led to anger; T-Rab/r, anger reaction; AX/In, anger in; AX/Out, anger out; AX/Contr, anger control; AX/EX, expression of anger; EQ, Empathy quotient; TAS-20, Toronto Alexithymia Scale-20 items; ERQ, The Emotion Regulation Questionnaire; Δ (%), Formula [−(T1 − T0)/T1] ∗ 100.

**Table 2 children-09-01418-t002:** Pearson’s correlation (*r*) between Toronto Alexithymia Scale (TAS-20) and psychological dimensions at baseline (T0; before the intervention) and T1 (after the intervention).

Psychological Dimensions	TAS-20 at T0		TAS-20 at T1	
	(*r*)	*p*-Values	(*r*)	*p*-Values
**STAXI** *S-Rab*	+0.248	**0.001**	−0.029	0.700
**STAXI** *T-Rab*	+0.247	**0.001**	−0.015	0.839
**STAXI** *T-Rab/t*	+0.119	0.122	+0.008	0.915
**STAXI** *T-Rab/r*	+0.126	0.100	+0.076	0.324
**STAXI** *AX/In*	+0.074	0.355	+0.041	0.590
**STAXI** *AX/Out*	+0.067	0.400	+0.069	0.372
**STAXI** *AX/Contr*	+0.916	0.008	+0.045	0.559
STAXI *AX/EX*	+0.080	0.314	+0.105	0.174
**ERQ** *Reappresal*	+0.043	0.584	+0.076	0.325
**ERQ** *Suppression*	+0.061	0.445	+0.133	0.080
**EQ**	−0.095	0.216	+0.005	0.944

Footnote: STAXI, State-trait anger expression inventory; S-Rab, state anger; T-Rab, trait anger; T-Rab/t, temperament led to anger; T-Rab/r, anger reaction; AX/In, anger in; AX/Out, anger out; AX/Contr, anger control; AX/EX, expression of anger; TAS-20, Toronto Alexithymia Scale-20 items; ERQ, The Emotion Regulation Questionnaire; EQ, Empathy Quotient.

**Table 3 children-09-01418-t003:** Reduction of TAS-20 score after the intervention (T1) among students with TAS-20 ≥ 61 at baseline (T0).

Measure	TAS-20 ≥ 61 at T0			TAS-20 ≥ 61 at T1			*t*	Changes (%) Total ΔT1-T0	*p*-Values
	Males (n = 20) (Mean ± SD)	Females (n = 39) (Mean ± SD)	Total (n = 59) (Mean ± SD)	Males (n = 20) (Mean ± SD)	Females (n = 39) (Mean ± SD)	Total (n = 59) (Mean ± SD)			
**TAS-20**	**67.4 ± 8.32**	**68.7 ± 5.14**	68.2 ± 6.51	40.7 ± 19.9	49.3 ± 15.4	46.0 ± 17.6	65.24	−48.2	**<0.0001**

**TAS-20**: Toronto Alexithymia Scale-20 items.

## Data Availability

Data are available from the authors upon request.

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
