# Peer review of "Psychoeducation Reduces Alexithymia and Modulates Anger Expression in a School Setting"

_children, 2022, doi:10.3390/children9091418_

Round 1

Reviewer 1 Report

Dear editor. Thank you very much for giving me the opportunity to review the article entitled “Psychoeducation reduces alexithymia and modulates anger expression in a school-setting”. The prevalence of alexithymia is high among students and occurs in conjunction with associated factors. Then the present work is relevant and necessary. In my opinion the work is well conducted. And the methodology is correct.

My criticisms to the authors are the following.

1.     1. If there are 6 meetings. Please can you describe more about each of them. And not only made mention of the them in a general way as they are in this report.

2.     2. I recommend including the flowchart of CONSORT

3.     3. The authors must show the implications of the results of their study.

Author Response

Dear Editor,

Many thanks for considering our paper for peer-review and providing the list of comments. My coauthors and I found reviewers’ suggestions very helpful and tried to revise the manuscript accordingly.

Here the answers to each comment and changes are tracked properly in the new version of manuscript.

A CONSORT flow-chart has been added as well as plagiarism-check performed,  even if the rate of repetitions was 28% in the first version (lower than 30%): most of the repetitions are related to tools names and standard description, not inter-changeable.

English has been also revised.

We really hope the MS is acceptable for publication in its current version on your venerable journal.

Best regards,

Antonio Ventriglio

#1

Dear editor. Thank you very much for giving me the opportunity to review the article entitled “Psychoeducation reduces alexithymia and modulates anger expression in a school-setting”. The prevalence of alexithymia is high among students and occurs in conjunction with associated factors. Then the present work is relevant and necessary. In my opinion the work is well conducted. And the methodology is correct.

My criticisms to the authors are the following.

  1. If there are 6 meetings. Please can you describe more about each of them. And not only made mention of the them in a general way as they are in this report.

R: Many thanks. The meetings have been described (a-e) and references about methods properly added.

  1. I recommend including the flowchart of CONSORT

R: CONSORT (2010) Flow Diagram has been provided in Figure 1 (attached) and mentioned in the manuscripe.

  1. The authors must show the implications of the results of their study.

R: a specific paragraph on possible implications of  the study findings has been added and it  improved the general take-home message.

Reviewer 2 Report

Dear editor and esteemed authors,

Thank you for giving me the opportunity to review the manuscript submitted to the journal Children. The article reports the results of a psychoeducational intervention aimed at reducing the levels of bullying in a sample of school children.

The results of the study suggest that the intervention evaluated significantly reduces the levels of alexithymia in the participants of both sexes, and the authors point out that the wording of this indicator and of other parameters examined can markedly reduce the levels of bullying in young adolescents, inside and outside the educational environment.

Considering that bullying is a global public health problem and that it negatively affects development during adolescence and in later stages, the evidence derived from this work should be taken into consideration by the designers of psychoeducational intervention programs to increase the effectiveness of these types of programs.

However, and despite the conceptual and methodological clarity of the work, it is necessary to complement and strengthen some parts of the text to improve its overall quality. Some of these elements are described below:

1. The theoretical introduction is excessively brief, and, although it mentions the results of a recent meta-analysis on the effectiveness of psychoeducational training programs in reducing bullying, the relationship between the different variables presented in the investigation.

2. It is recommended to consult the results of both systematic reviews and meta-analyses that describe the associations between the variables examined in the study.

3. From my point of view, the description of the intervention is too brief and in some cases imprecise. Taking into account that the strong point of the manuscript is precisely to show the effectiveness of the program in reducing bullying, the description of the activities and the intervention itself should be broader.

4. Given the important role played by the context (educational in this case) in the implementation of intervention programs, it would be advisable to offer a more detailed description of the educational context, and even of the social context in which the educational center is located and the intervention takes place.

5. Explain in detail the participant selection process.

6. Include a limitations section (in the discussion).

7. I believe that is important to separate de disussion and the conclusion section and to center the discussion to explain how the resultas of this intervention could serve to improve the design, the content and the implementation of psychoeducational programs to prevent bullying in school settings.

Best regards,

Author Response

Dear Editor,

Many thanks for considering our paper for peer-review and providing the list of comments. My coauthors and I found reviewers’ suggestions very helpful and tried to revise the manuscript accordingly.

Here the answers to each comment and changes are tracked properly in the new version of manuscript.

A CONSORT flow-chart has been added as well as plagiarism-check performed,  even if the rate of repetitions was 28% in the first version (lower than 30%): most of the repetitions are related to tools names and standard description, not inter-changeable.

We really hope the MS is acceptable for publication in its current version on your venerable journal.

Best regards,

Antonio Ventriglio

#2

Dear editor and esteemed authors,

Thank you for giving me the opportunity to review the manuscript submitted to the journal Children. The article reports the results of a psychoeducational intervention aimed at reducing the levels of bullying in a sample of school children.

The results of the study suggest that the intervention evaluated significantly reduces the levels of alexithymia in the participants of both sexes, and the authors point out that the wording of this indicator and of other parameters examined can markedly reduce the levels of bullying in young adolescents, inside and outside the educational environment.

Considering that bullying is a global public health problem and that it negatively affects development during adolescence and in later stages, the evidence derived from this work should be taken into consideration by the designers of psychoeducational intervention programs to increase the effectiveness of these types of programs.

However, and despite the conceptual and methodological clarity of the work, it is necessary to complement and strengthen some parts of the text to improve its overall quality. Some of these elements are described below:

  1. The theoretical introduction is excessively brief, and, although it mentions the results of a recent meta-analysis on the effectiveness of psychoeducational training programs in reducing bullying, the relationship between the different variables presented in the investigation.

R: The introduction has been developed with more literature on the topic (with no aim to systematically review it) and new recent  references considered

  1. It is recommended to consult the results of both systematic reviews and meta-analyses that describe the associations between the variables examined in the study.

R: Results from significant studies re. the topic have been reported in the introduction

  1. From my point of view, the description of the intervention is too brief and in some cases imprecise. Taking into account that the strong point of the manuscript is precisely to show the effectiveness of the program in reducing bullying, the description of the activities and the intervention itself should be broader.

R: Many thanks. The modules of the intervention (meetings) have been described (a-e) and references about methods properly added.

  1. Given the important role played by the context (educational in this case) in the implementation of intervention programs, it would be advisable to offer a more detailed description of the educational context, and even of the social context in which the educational center is located and the intervention takes place.

R: Many thanks for this interesting point. More details about reasons for  this psychoeducational initiative as well as on the educational context have been added in the methods

  1. Explain in detail the participant selection process.

R: CONSORT (2010) Flow Diagram has been provided in Figure 1 (attached) and mentioned in the manuscript

  1. Include a limitations section (in the discussion).

R: limitations of study-design and findings have been discussed and implemented

  1. I believe that is important to separate de disussion and the conclusion section and to center the discussion to explain how the resultas of this intervention could serve to improve the design, the content and the implementation of psychoeducational programs to prevent bullying in school settings.

R: Conclusions have been separated as suggested and  a specific paragraph on possible implications of  the study findings has been added for improving  the general take-home message  

Round 2

Reviewer 1 Report

The authors have addressed all the criticisms made by this reviewer, and

corrected the text accordingly. So the article is much more understandable and

shows the significance of the results. Thus, I consider the article accepted for publication in this journal.

Author Response

Answers to reviewers

Dear Editor,

Many thanks for providing this positive report and suggesting minor revisions.

My coauthors and I addressed comments of reviewer  #2 (yellow changes).

We really hope the MS is acceptable for publication in its current version on your venerable journal.

Best regards,

Antonio Ventriglio

Reviewer 2 Report

Dear authors,

Thank you for giving me the opportunity to review the revised version of the manuscript. After reviewing the changes introduced, I consider that the manuscript has gained in internal consistency and has notably increased its overall merit. However, I believe that the authors miss the opportunity to show in detail the practical applications of the research results, to inform the design of intervention programs to reduce aggressive behavior in the school environment.

Regards,

Author Response

(The authors gave the same response as above.)
